# Radiomic Signatures Derived from Hybrid Contrast-Enhanced Ultrasound Images (CEUS) for the Assessment of Histological Characteristics of Breast Cancer: A Pilot Study

**DOI:** 10.3390/cancers14163905

**Published:** 2022-08-12

**Authors:** Ioana Bene, Anca Ileana Ciurea, Cristiana Augusta Ciortea, Paul Andrei Ștefan, Larisa Dorina Ciule, Roxana Adelina Lupean, Sorin Marian Dudea

**Affiliations:** 1Department of Radiology, “Iuliu Hatieganu” University of Medicine and Pharmacy, 400012 Cluj-Napoca, Romania; 2Department of Radiology, Emergency County Hospital, 400006 Cluj-Napoca, Romania; 3Anatomy and Embryology, Morphological Sciences Department, “Iuliu Hațieganu” University of Medicine and Pharmacy, Victor Babes, Street, Number 8, 400012 Cluj-Napoca, Romania; 4Department of Biomedical Imaging and Image-Guided Therapy, General Hospital of Vienna (AKH), Medical University of Vienna, 1090 Vienna, Austria; 5Department of Oncology, Emergency County Hospital, 400006 Cluj-Napoca, Romania; 6Histology, Morphological Sciences Department, “Iuliu Hațieganu” University of Medicine and Pharmacy, Louis Pasteur Street, Number 4, 400349 Cluj-Napoca, Romania

**Keywords:** contrast-enhanced ultrasound, hybrid images, radiomic analysis, breast cancer

## Abstract

**Simple Summary:**

The hybrid images obtained at the CEUS present an overlap of the tumor in grayscale over the enhancement pattern, which allows an appropriate evaluation of the entire lesion without including in the analysis the peritumoral area. We hypothesized that the spatial heterogeneity of tissue enhancement differs depending on histological characteristics. Therefore, the aim of this study was to evaluate the diagnostic performance of radiomic features extracted from standardized hybrid CEUS data for the assessment of hormone receptor status, HER2 status, tumor grade and Ki-67 in patients with primary breast cancer. The clinical benefit of this study would be to use CEUS and bioinformatics tools for improved biopsy planning or even guiding treatment decisions.

**Abstract:**

The purpose of this study was to evaluate the diagnostic performance of radiomic features extracted from standardized hybrid contrast-enhanced ultrasound (CEUS) data for the assessment of hormone receptor status, human epidermal growth factor receptor 2 (HER2) status, tumor grade and Ki-67 in patients with primary breast cancer. Methods: This prospective study included 72 patients with biopsy-proven breast cancer who underwent CEUS examinations between October 2020 and September 2021. Results: A radiomic analysis found the WavEnHH_s_4 parameter as an independent predictor associated with the HER2+ status with 76.92% sensitivity, and 64.41% specificity and a prediction model that could differentiate between the HER2 entities with 76.92% sensitivity and 84.75% specificity. The RWavEnLH_s-4 parameter was an independent predictor for estrogen receptor (ER) status with 55.93% sensitivity and 84.62% specificity, while a prediction model (RPerc01, RPerc10 and RWavEnLH_s_4) could differentiate between the progesterone receptor (PR) status with 44.74% sensitivity and 88.24% specificity. No texture parameter showed statistically significant results at the univariate analysis when comparing the Nottingham grade and the Ki-67 status. Conclusion: Our preliminary data indicate a potential that hybrid CEUS radiomic features allow the discrimination between breast cancers of different receptor and HER2 statuses with high specificity. Hybrid CEUS radiomic features might have the potential to provide a noninvasive, easily accessible and contrast-agent-safe method to assess tumor biology before and during treatment.

## 1. Introduction

Contrast-enhanced ultrasound (CEUS) is a valuable imaging technique in the assessment of breast pathology. Several studies in the literature have demonstrated the usefulness of CEUS in differentiating benign and malignant breast lesions, evaluating pathological characteristics of breast cancer (estrogen and progesterone receptors, histological grade, human epidermal growth factor receptor 2, Ki-67), detecting the sentinel lymph node and even predicting the response to neoadjuvant chemotherapy (NAC) [1,2,3,4].

Breast tumor structure consists of a high degree of heterogeneity observed at histological and genetic levels, and increased intratumor genetic heterogeneity is associated with poor clinical outcomes.

Radiomics (through texture analysis) is a technique based on the extracting and processing of image-specific parameters that provide an objective description of image content by quantifying the distribution patterns and intensity of pixels. Radiomics is able to extract features from medical images that can be used to understand relationships between the imaging heterogeneity and the genetic and phenotypical characteristics of tumors, and it can be integrated in prediction models that may improve diagnostic, prognostic and even expect treatment outcomes. The basic principle of the ultrasonography-based radiomic analysis is that a pathological process that alters the tissue produces a modified ultrasound signal, which will in turn give textural features with different values from those of the normal structure [5,6,7].

Radiomics allows an objective and quantitative characterization of the morphology, texture and pharmacokinetic behavior of breast tumors by obtaining features that are not visible to the naked eye, therefore improving the predictive power of medical imaging [8].

Breast radiomics studies are mostly applied to the diagnosis of breast cancer, the prediction of the molecular classification, lymph nodes metastasis and molecular markers of invasiveness [9,10,11,12]. Luo et al. [13] found in their study that radiomics had better performance in distinguishing breast lesions than BI-RADS, and according to Wang et al. [14], radiomics was superior in differentiating small benign and malignant breast lesions.

Several studies have already reported the value of radiomic analyses of contrast-enhanced imaging techniques such as dynamic contrast-enhanced magnetic resonance imaging (DCE-MRI) and contrast-enhanced spectral mammography (CESM) in predicting breast cancer molecular subtypes with a similar performance [9,10,15,16]. Both techniques provide functional information on neoplastic neo-angiogenesis reflecting underlying tumor biological characteristics. Furthermore, Wang et al. [17] found a correlation between CEUS and acoustic radiation force impulse (ARFI), which also provides quantitative features, regarding the assessment of HER-2 expression levels in breast cancer.

The hybrid images obtained at the CEUS represent an overlap of the tumor in grayscale over the enhancement pattern, which allows an appropriate evaluation of the entire lesion without including in the analysis the peritumoral area.

We hypothesized that the spatial heterogeneity of tissue enhancement differs depending on histological characteristics. Therefore, the aim of this study was to evaluate the diagnostic performance of radiomic features extracted from standardized hybrid CEUS data for the assessment of hormone receptor status, HER2 status, tumor grade and Ki-67 in patients with primary breast cancer.

The clinical benefit of this study would be to use CEUS and bioinformatics tools for improved biopsy planning or even guiding treatment decisions.

## 2. Materials and Methods

This prospective study was approved by the Institutional Review Board (Number/Date: 280/11 August 2020), and informed consent was obtained from each patient before performing CEUS. The informed consent included information related to the purpose and duration of the study; risks; benefits; responsibility of the participant; the rights of the participant to withdraw from the study; confidentiality; and the procedure, including the way of administration of the contrast agent and contraindications.

### 2.1. Study Design and Population

Between October 2020 and September 2021, 87 patients underwent CEUS examinations.

CEUS was performed only in patients with suspicious breast lesions detected at mammography and/or ultrasound and classified according to the American College of Radiology (ACR) Breast Imaging-Reporting and Data System (BI-RADS) in 4A, 4B, 4C or 5. None of the included patients had contraindications for the administration of contrast media such as hypersensitivity to sulfur hexafluoride (or any of the components of SonoVue^®^), acute cardiac failure, recent acute coronary syndrome or clinically unstable ischemic cardiac disease, known right-to-left shunts, severe rhythm disorders or pulmonary hypertension (pulmonary artery pressure >90 mmHg), uncontrolled systemic hypertension and respiratory distress syndrome.

All the patients agreed to perform biopsy.

We excluded from the study benign biopsy-proven lesions and malignant lesions under 6 mm diameter due to the difficulty of depicting their margins.

Figure 1 summarizes the flowchart of the patient selection.

### 2.2. Image Acquisition

All breast CEUS examinations were acquired with a LOGIQ S8 ultrasound machine (General Electric Ultrasound Korea, Seongnam-si, Korea).

Before the contrast examination, the lesion was assessed on both grayscale and color Doppler ultrasound to identify the scanning plane with the largest diameter of the lesion and with the richest vascularization.

For all the examinations, we set the mechanical index (MI) at 0.06, the gain of 100–120 dB, single focus, image depth of approximately 3–4 cm and SonoVue^®^ (sodium hexachloride microbubbles; Bracco, Amsterdam, The Netherlands) was administered intravenously as a bolus of 2.4 mL followed by a flush of 5 mL sodium chloride 0.9% using a 20-gauge cannula.

Before the examination, we asked the patients not to speak or move and avoid coughing or extensive breathing movements during the image acquisition in order to avoid artifacts.

All CEUS examinations were performed by the same physician, before the biopsy and the images were recorded for at least 180 s. We started recording the examination synchronously with the injection of the contrast agent. No patient presented side effects to SonoVue^®^ (sensitivity at the injection site, hypersensitivity reactions such as facial flushing, nausea, bradycardia, hypotension or anaphylactic shock).

### 2.3. Image Processing

The images were processed by the same physician who performed the examinations using the time-intensity curve (TIC) analysis software integrated into the ultrasound device.

The start cine loop was selected when the first microbubbles arrived in the lesion, and the stop cine loop at the end of the examination (around second 180). On the touch panel for the fitting curve, “Gamma variate” was selected, and as smoothing “Gaussian 600 ms”.

A region of interest (ROI) was placed in the most perfused area of the malignant lesion. Corresponding to the peak of the selected ROI, the “Hybrid image” option was selected on the touch panel.

### 2.4. Reference Standard

The pathology reports of all biopsied lesions were considered as the gold standard. The reports of malignant lesions included information regarding the histological tumor grade, the estrogen receptor (ER), progesterone receptor (PR), and human epidermal growth factor receptor 2 (HER2) status and the value of Ki-67.

### 2.5. Texture Analysis (TA)

A texture analysis was conducted on hybrid CEUS images acquired at the peak. In the MaZda program, all the hybrid images had a black and white appearance, but even in these conditions the overlap of the perfusion map on the grayscale image could be observed even with the naked eye.

The TA protocol comprised of the following five steps: image pre-processing, lesion segmentation, feature extraction, feature selection and prediction.

To ensure the repeatability and the reproducibility of our results, in the first step of our texture analysis process, the images were normalized. Therefore, we excluded the grey-level variations that could affect the true textures of the images. In the second step, the segmentation process was performed semi-automatically, based on gradient and geometry coordinates, therefore lowering the variability produced by the manual segmentation.

#### 2.5.1. Image Pre-Processing and Segmentation

Tumor segmentation was performed by an experienced breast radiologist. Contours were depicted on hybrid images. Multifocality or bilaterality of breast tumors was not observed in any patient.

A semi-automatic level-set technique was used for the definition and positioning of ROI using gradient and geometry coordinates. As this technique does not require the manual delineation of the structure of interest contours, the inter- and intra-observer reproducibility was not assessed in this study. The researcher placed a seed in the area of interest, and the software automatically delineated the area based on gradient and geometrical contours. When necessary, manual corrections were applied (Figure 2). Before the extraction of texture parameters, the imported image’s grey levels were normalized based on the mean and three standard deviations of grey level intensities to reduce the contrast and brightness variations (which could affect the true image textures). The parameters are detailed in the table below (Table 1).

#### 2.5.2. Feature Extraction

The feature computation from every ROI was automatically performed by the MaZda software version 4.6 (MaZda, Institute of Electronics, Technical University of Lodz, Lodz, Poland).

#### 2.5.3. Feature Selection

First, the MaZda software allows the selection of the best-suited parameters for differentiating between classes through several reduction methods. We applied a selection method based on Fisher coefficients (F, the ratio of between-class to within-class variance). This method allowed us to select a set of the ten most discriminative features that were best suited to discriminate between selected groups [18].

Second, the parameters highlighted by the selection method underwent a statistical analysis. The absolute values recorded by each of the previously selected parameters were compared between groups by conducting a univariate analysis test (the Mann–Whitney U test). The statistical significance level was set at a *p*-value of below 0.05. Features that did not meet the above-mentioned criteria were excluded from further analysis.

Abbreviations of feature characteristics produced by the extraction algorithm appeared in the feature name generated by the MaZda software. The first letter indicated the color channel through which the image was computed (R” identifies the red color channel). This color channel was automatically chosen by the software; the investigators had no intervention on this procedure.

#### 2.5.4. Class Prediction

A multivariate analysis was performed to identify which of the previously selected parameters were independent predictors for PR and HER2 status. The analysis was constructed using the “enter” input model, and the coefficient of determination (R-squared) and the variance inflation factor (VIF) were computed from the same data. The final prediction model included only variables that showed a *p*-value of below 0.05 following a multivariate analysis. In addition, features that showed high VIF (>10^4^) were excluded from further analysis due to multicollinearity. Finally, the predicted values were saved and subsequently used in a receiver operating characteristics (ROC) analysis to assess the diagnostic power of the entire prediction model. The ROC analysis was also used to determine the diagnostic power of other parameters that showed statistically significant results at the univariate analysis. The ROC analysis implied the calculation of the area under the curve (AUC), sensitivity and specificity, all with 95% confidence intervals (CIs). Sensitivity (Se) and specificity (Sp) were computed from the same data, without further adjustments. A statistical analysis was performed using a commercially available dedicated software, MedCalc v14.8.1 (MedCalc Software, Mariakerke, Belgium).

## 3. Results

A total of 72 patients with biopsy-proven malignant breast lesions were included in the study. Participants’ ages ranged from 29 to 83 years (56.5 + 14.34, mean age + SD). All the characteristics regarding the patients are listed in Table 2.

The extracted parameters for each histological characteristic (Nottingham grade, ER, PR, HER2 status and Ki-67) are listed in Table 3.

The multivariate analysis involving the HER2 status showed a coefficient of determination of 0.2, an R2-adjusted of 0.12, MCC of 0.45 and a residual standard deviation of 0.36. One parameter (WavEnHH_s_4) was independently associated with the HER2+ status (Table 4).

The ROC analysis of the WavEnHH_s_4 parameter showed a statistical significance level of 0.003, an AUC of 0.726 (CI, 0.608–0.825) and a Youden index of 0.4133. For a cut-off value of ≤8.124, the WavEnHH_s_4 parameter could differentiate between the HER2 entities with 76.92% Se (CI, 46.2–95%), and 64.41% Sp (CI, 50.9–76.4%). For the same scope, the ROC analysis of the HER2 prediction model showed a statistical significance level of <0.0001, and an AUC of 0.836 (CI, 0.73–0.913) and a Youden index of 0.6167. For a cut-off value of >0.2840, the prediction model could differentiate between the HER2 entities with 76.92% Se (CI, 46.2–95%) and 84.75% Sp (CI, 73.0–92.8%) (Figure 3).

Table 5 shows the diagnostic values of the parameters that were included in the prediction model but were not independently associated with the HER2 status.

The multivariate analysis involving the PR status showed a coefficient of determination of 0.97, R2-adjusted of 0.5, MCC of 0.31 and a residual standard deviation of 0.48. Neither of the three included parameters were independent predictors for the PR+ status (Table 6).

The ROC analysis of the prediction model that incorporated all three parameters that showed statistically significant results at the univariate analysis (RPerc01, RPerc10 and RWavEnLH_s_4) showed a significance level of 0.0048, and an AUC of 0.678 (CI, 0.554–0.802) and a Youden index of 0.3297. For a cut-off value of >0.617, the prediction model could differentiate between the PR status with 44.74% Se (CI, 28.6–61.7%) and 88.24% Sp (CI, 62.1–91.3%) (Figure 4).

No texture parameter showed statistically significant results at the univariate analysis when comparing the Nottingham grade and the Ki-67 status.

Another wavelet transformation parameter (RWavEnLH_s-4) was the only one that showed statistically significant results when comparing the ER+ and ER− patients (*p* = 0.0181). The ROC analysis for this parameter in the diagnosis of ER+ patients showed a significance level of 0.0015, and an AUC of 0.711 (CI, 0.592–0.811) and a Youden index of 0.4. For a cut-off value of ≤32.81, the RWavEnLH_s-4 parameter was able to identify ER+ tumors with 55.93% (CI, 42.4–68.8%) sensitivity and 84.62% (CI, 54.6–98.1%) specificity (Figure 5).

## 4. Discussion

In this study, we investigated the diagnostic performance of radiomic features derived from hybrid images obtained at CEUS for the assessment of breast cancer receptor status, HER2 and Ki-67 status and histological grade. We hypothesized that the spatial heterogeneity of tissue enhancement differs depending on histological characteristics and could be quantified by a radiomic analysis. Our results demonstrate that hybrid CEUS radiomic features allow the discrimination between breast cancers of different receptor and HER2 statuses with high specificity.

To the best of our knowledge, no study has been conducted to investigate the usefulness of a radiomic analysis of hybrid CEUS images in order to predict the hormone receptor status, HER2 status, tumor grade and Ki-67 in patients with primary breast cancer. Currently, the way to determine these prognostic factors is through invasive tissue sampling; however, it may not be representative of tumor biology in its entirety, with the process depending on site selection or sampling bias. Therefore, there is a strong argument for the development of alternatives to the formal biopsy analysis that could provide the information which reflects the tumor in its entirety; thus, the performance of radiomics analysis is worth being investigated.

Wavelet transformation is a multiresolution technique for transforming images into representations that include both spatial and frequency information [19]. This transformation allows for the quantification of an image’s frequency content, which is proportional to the image’s gray level variations. To begin, images are scaled five times in both vertical and horizontal directions. In addition, to separate the image data, two types of filters (high and low pass) are applied [20]. Finally, different subbands are extracted from an image divided into four parts, each of which corresponds to a different frequency component. The result is a five-scaled image with four frequency bands on every scale. Each subband’s energy feature is calculated. The distribution of this energy along the frequency axis over scale and orientation is quantified through the wavelet energy parameter, which reflects the local uniformity of an image. When an image’s grey levels are distributed in a consistent or periodic pattern, energy levels become high [19]. We obtained higher values of the WavEnHH_s_4 parameter for the HER2− than for the HER2+ patients, meaning that HER2+ tumors have an inhomogeneous distribution of gray levels within the pixels. This aspect was consistent with the literature, according to which HER2+ tumors have a higher proliferation rate, areas of necrosis and fibrosis leading to an inhomogeneous appearance on DCE MRI and CEUS [21,22]. Furthermore, La Forgia et al. [23] concluded that a radiomics analysis of CESM images can distinguish HER2 positive and triple-negative breast cancers. It would be interesting to conduct a study in order to investigate if this parameter could predict the HER2+ status in benign lesions, thus producing false positive results.

Wu et al. [11] conducted a study using a radiomics analysis on grayscale ultrasound images that managed to predict the HER2 and hormone receptor status of DCIS with good accuracy. In our study, the multivariate analysis showed that no parameters were independent predictors for the PR+ status. Again, a wavelet energy parameter (RWavEnLH_s_4) showed statistically significant results at the univariate analysis and held higher values for the PR− than for PR+ breast tumors. In addition, two percentile parameters also showed statistically significant results when comparing the PR+ and PR− patients. The percentile value (*n*) is the point at which n% of the pixel values that form the histogram are found to the left [24]. In other words, a percentile gives the highest gray-level value under which a given percentage of the pixels in the image is contained [20]. This signifies that 1% and 10% of the pixels within images were distributed under higher values for PR− than for PR+ patients.

When comparing ER+ and ER− patients, a single wavelet transformation parameter (RWavEnLH_s-4) showed statistically significant results. This parameter held higher values for ER− than for ER+ patients. Our results were contrary to prior studies, or what we expected, because breast cancers with positive estrogen receptors have a lower rate of cell proliferation, so less necrotic tissue, and a homogeneous appearance at CEUS. On the other hand, breast cancers with negative ER present areas of central necrosis and fibrosis, and therefore an inhomogeneous appearance at CEUS [21,25,26,27,28]. However, in these articles, the lesion enhancement pattern was assessed subjectively, compared to the objective evaluation performed through a radiomic analysis. Furthermore, in the study, we did not take into account the entire tumor biology (tumor cellularity, areas of hyaline stroma, tumor-infiltrating mononuclear lymphoid cells, areas of necrosis and microcalcifications) that might influence this parameter.

Our study showed no texture parameter with statistically significant results at the univariate analysis when comparing the Ki-67 status and the Nottingham grade. Ki-67 expression indicates a guiding role in evaluating the efficacy of neoadjuvant chemotherapy for breast cancer, being closely related to the patient’s prognosis [29]. Demircioglu et al. [30] constructed radiomics models for predicting Ki67 expression in invasive breast cancer based on eight features extracted from MRI images, with an AUC of 0.81.

Marino et al. [31] investigated the potential of a radiomic analysis of both DCE-MRI and CESM of the breast for predicting tumor invasiveness, hormone receptor status (HR) and tumor grade in patients with primary breast cancer. They included 48 female patients with 49 biopsy-proven breast cancers. An MRI radiomic analysis yielded classification accuracies of up to 90% for invasive vs. noninvasive breast cancers, 82.6% for HR+ vs. HR− breast cancers and 77.8% for G1 + G2 vs. G3 cancers. A CESM radiomic analysis yielded classification accuracies of up to 92% for invasive vs. noninvasive breast cancers, 95.6% for HR+ vs. HR− breast cancers and 77.8% for G1 + G2 vs. G3 invasive cancers. Furthermore, in a radiomic analysis, Bhooshan et al. [32] also found that DCE-MRI was promising for distinguishing breast tumors of different grades.

CEUS is an imaging technique comparable in terms of cost and performance time to CESM, but it involves less time and lower costs compared to DCE-MRI. This new approach, which involves the radiomic analysis of hybrid CEUS images, has as additional costs to CEUS, the costs related to the MaZda program. Furthermore, it would be interesting to conduct a study in order to explore the learning curve of this method, although we predict that it is similar to the ones involving other imaging techniques.

Despite some significant findings, the work presented has limitations. First, it is a single institution prospective study, and the sample size is small, limiting accurate analysis. It would be interesting to evaluate a larger database, with examinations performed in several institutions, even if slightly different CEUS imaging protocols were used, to compare the results and evaluate the reproducibility of the method. Second, some molecular subtypes (HER2+ and triple-negative) are too poorly represented. Third, the relatively low sensitivity compared to conventional existing tests is a major limitation of our method. However, this method could still be used as an additional criterion to increase diagnostic confidence, especially since the technique required for the extraction of the TA parameters is relatively straightforward. Furthermore, all CEUS examinations were performed by a single radiologist, which eliminated inter-operator variability, but we used the same parameters for every examination, and thus by following the protocol the method could be reproducible. Finally, it would be of interest to perform an intra-tumoral analysis based on other imaging modalities, such as DCE-MRI, to validate and extend this work. Therefore, additional work is needed before these methods can be utilized to facilitate the noninvasive assessment of breast cancer histopathological characteristics in clinical practice.

## 5. Conclusions

Our preliminary data indicate a potential that hybrid CEUS radiomic features allow the discrimination between breast cancers of different receptor and HER2 statuses with high specificity. Although hybrid CEUS radiomic features are unlikely to replace formal tissue biopsy and analysis, they might have the potential to provide a noninvasive, easily accessible and contrast-agent-safe method to assess tumor biology before and during treatment.

## Figures and Tables

**Figure 1 cancers-14-03905-f001:**
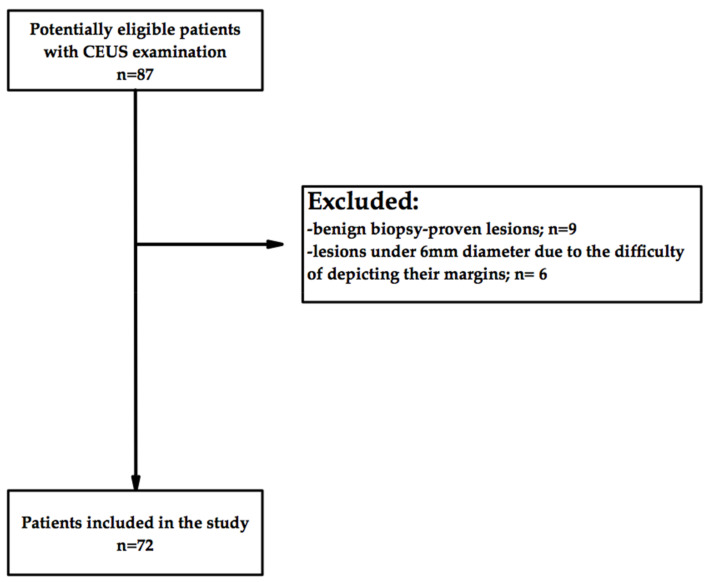
Flowchart of patient inclusion in the study.

**Figure 2 cancers-14-03905-f002:**
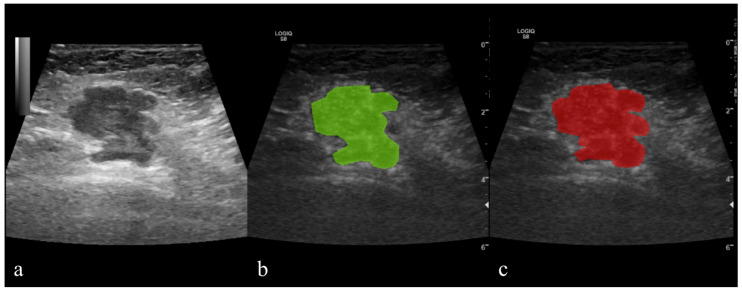
Contrast-enhanced ultrasound (CEUS) of histologically proven breast cancer; (**a**) grayscale image; (**b**,**c**) hybrid images with the tumor applied regions of interest (ROIs) that were automatically delineated by the software (green) and the final ROI after manual correction (red).

**Figure 3 cancers-14-03905-f003:**
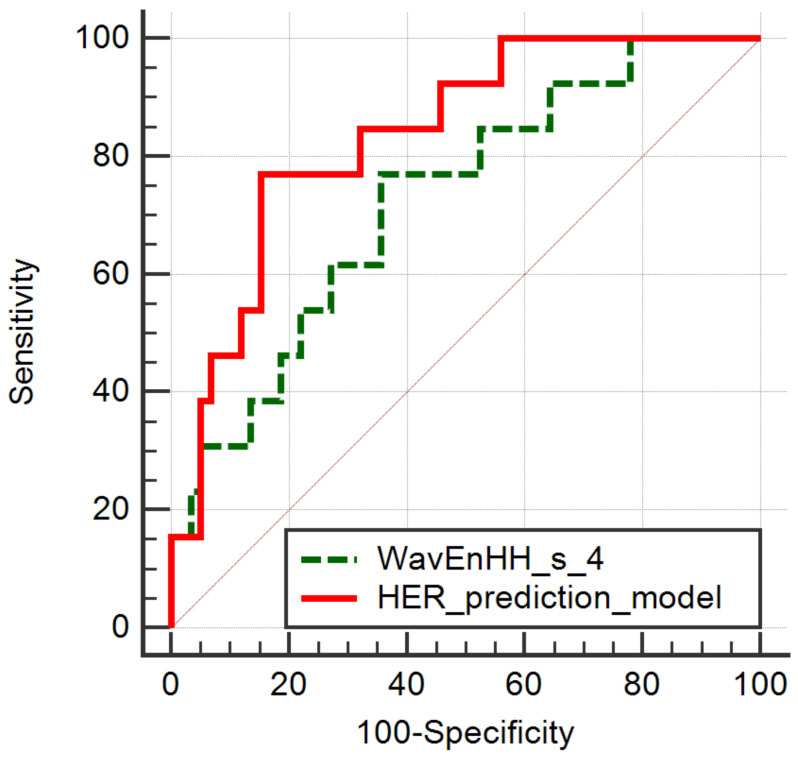
ROC curves of the WavEnHH_s_4 parameter that was independently associated with the HER+ status and the HER prediction model.

**Figure 4 cancers-14-03905-f004:**
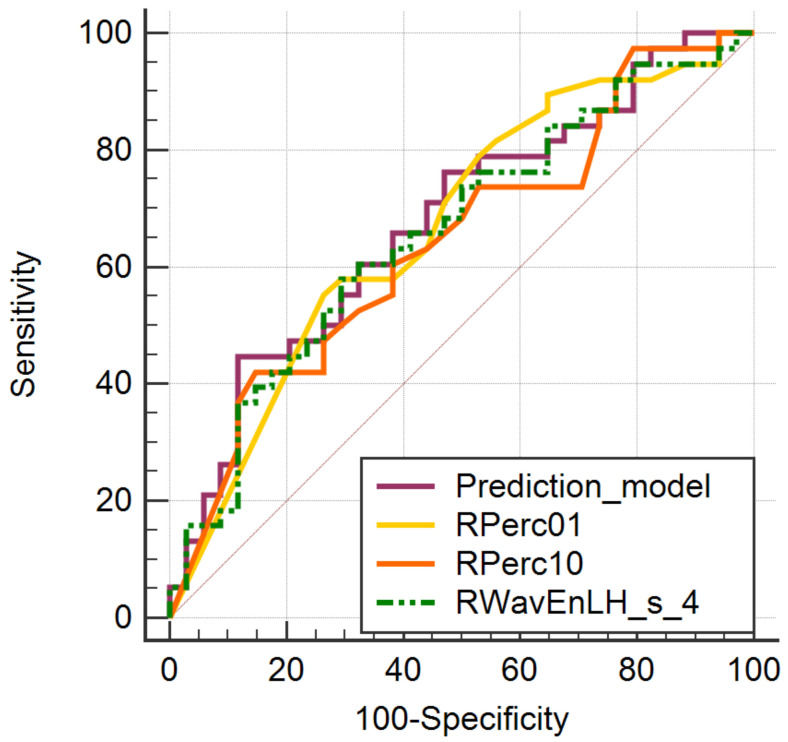
ROC curves of the parameters that showed statistically significant results in the comparison of PR+ and the PR− patients (RPerc01, RPerc10 and RWavEnLH_s_4) and of the prediction model that incorporated the three parameters.

**Figure 5 cancers-14-03905-f005:**
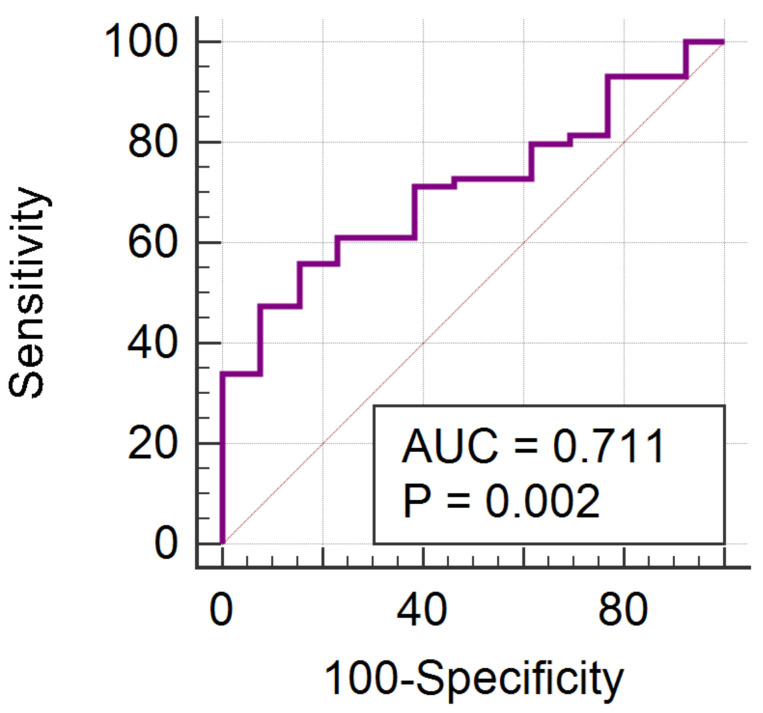
ROC curve of the RWavEnLH_s-4 in the diagnosis of ER+ tumors.

**Table 1 cancers-14-03905-t001:** Texture parameters.

Radiomics Features	Computation Method	Computational Variations	Class
Perc.01–99%, Skewness, Kurtosis, Variance, Mean	**-**	**-**	Histogram
GLevNonU, LngREmph, RLNonUni, ShrtREmp, Fraction	6 bits/pixel	4 directions	RLM
Teta 1–4, Sigma	-	-	ARM
InvDfMom, SumAverg, SumVarnc, SumEntrp, Entropy, DifVarnc, DifEntrp, AngScMom, Contrast, Correlat, SumOfSqs	6 bits/pixel; 5 between-pixel distances	4 directions	COM
WavEn	5 scales	4 frequency bands	WT
GrNonZeros, percentage of pixels with nonzero gradient, GrMean, GrVariance, GrSkewness, GrKurtosis	4 bits/pixel	-	AR

AR, absolute gradient; RLM, run length matrix; COM, co-occurrence matrix; ARM, auto-regressive model; WT, wavelet transformation; Mean, histogram’s mean; Variance, histogram’s variance; Skewness, histogram’s skewness; Kurtosis, histogram’s kurtosis; Perc.01–99%, 1st to 99th percentile; GrMean, absolute gradient mean; GrVariance, absolute gradient variance; GrSkewness, absolute gradient skewness; GrKurtosis, absolute gradient kurtosis; GrNonZeros, percentage of pixels with nonzero gradient; RLNonUni, run-length nonuniformity; GLevNonU, grey level nonuniformity; LngREmph, long-run emphasis; ShrtREmp, short-run emphasis; Fraction, the fraction of image in runs; AngScMom, angular second moment; Contrast, contrast; Correlat, correlation; SumOfSqs, the sum of squares; InvDfMom, inverse difference moment; SumAverg, sum average; SumVarnc, sum variance; SumEntrp, sum entropy; Entropy, entropy; DifVarnc, the difference of variance; DifEntrp, the difference of entropy; Teta 1–4, parameters θ1–θ14; Sigma, parameter σ; WavEn, wavelet energy.

**Table 2 cancers-14-03905-t002:** Patients’ characteristics.

Patients’ Characteristics	Number	Frequency (%)
**Mean age, years (range)**	56.5 (29–83)	
**Mean lesion size, mm (range)**	24.07 (7–60)	
**Pathological Type**		
DCIS	1	1.38
DCIS + Borderline phyllodes	1	1.38
IDC NST	63	87.56
IDC with mucinous components	1	1.38
Tubular carcinoma	1	1.38
Papillary carcinoma	2	2.77
Malignant phyllodes	1	1.38
ILC	2	2.77
**Nottingham grade**		
I	11	15.28
II	39	54.17
III	22	30.55
**Estrogen receptor status**		
Negative	13	18.05
Positive	59	81.95
**Progesterone receptor status**		
Negative	34	40.27
Positive	38	59.73
**Her2 status**		
Negative	59	81.94
Positive	13	18.06
**Ki-67**		
<20%	35	48.61
≥20%	37	51.39
**Luminal subtype**		
A	29	40.27.
B	30	41.66
HER2+	6	8.33
Triple-negative	7	9.74

**Table 3 cancers-14-03905-t003:** Texture parameter depending on histological characteristics.

**Nottingham I vs. Nottingham II + III**						
**Texture Parameters**	**F**	***p*-Value**	**Nottingham I**		**Nottingham II + III**	
			**Median**	**IQR**	**Median**	**IQR**
RATeta3	0.3089	0.2832	0.49	0.43–0.56	0.51	0.46–0.57
RATeta4	0.2881	0.3289	0.16	0.12–0.17	0.14	0.12–0.16
RCV4D6DifVarnc	0.2458	0.2293	6.53	3.37–7.71	7.04	5.51–7.98
RCV3D6DifVarnc	0.2428	0.321	5.15	2.80–6	5.45	4.42–6.07
RCV5D6DifVarnc	0.238	0.2356	7.43	3.77–9.16	8.39	6.18–9.69
RCV2D6DifVarnc	0.2191	0.4151	3.45	2–3.91	3.52	2.89–3.91
RWavEnHH_s-4	0.2097	0.1532	6.44	4.18–10.63	9.67	6.40–13.32
RATeta2	0.2073	0.3132	−0.58	−0.64–−0.48	−0.59	−0.64–−0.55
RCZ3D6DifVarnc	0.2029	0.3535	6.52	3.23–7.06	6.47	5.34–7.42
RCN2D6DifVarnc	0.2006	0.4244	3.71	2.24–4.75	4.07	3.31–4.71
**ER Negative vs. ER Positive**						
**Texture Parameters**	**F**	***p*-Value**	**ER Negative**		**ER Positive**	
			**Median**	**IQR**	**Median**	**IQR**
RWavEnLH_s-4	0.4951	0.0181	39.44	34.77–51.97	31.64	18.5–44.76
RWavEnHL_s-4	0.3413	0.0854	26.66	17.677–31.83	18.34	11.67–27.36
RATeta1	0.3017	0.1535	0.93	0.92–0.94	0.93	0.91–0.93
RWavEnLL_s-6	0.2874	0.1665	1511.37	1024.61–2972.41	1168.38	711.04–1775.91
RWavEnLL_s-5	0.2851	0.1901	1347.99	1034.33–3232.67	1137.51	703.89–1882.82
RWavEnHL_s-6	0.2467	0.1535	25.66	14.75–46.63	16.93	12.07–30.72
RPerc10	0.2042	0.1141	15	7.5–25.75	10	3–15.75
RWavEnLL_s-4	0.1957	0.2627	1193.40	889.87–3281.36	1108	688.71–1738.54
RPerc50	0.1926	0.2951	25	21.5–53.25	24	18.25–35
RWavEnLL_s-3	0.1771	0.3758	1238.59	744.08–3315.79	1019.02	676.08–1729.19
**PR Negative vs. PR Positive**						
**Texture Parameters**	**F**	***p*-Value**	**PR Negative**		**PR Positive**	
			**Median**	**IQR**	**Median**	**IQR**
RPerc01	0.3341	0.0108	7	2–12	2	2–7
RPerc10	0.325	0.0352	13.5	6–20	8.5	2–16
RWavEnLH_s-4	0.2964	0.0195	36.66	27.74–48.28	28.88	17.57–38.36
RWavEnLL_s-5	0.2809	0.1117	1408.29	773.71–2509.74	1125.22	617.94–1697.62
RWavEnLL_s-4	0.2634	0.1307	1257.85	729.63–2883.87	1039.67	533.92–1671.16
RPerc50	0.2541	0.1181	25	21–43	23.5	15–33
RWavEnLL_s-3	0.2473	0.152	1324.88	715.54–2705.42	1008.67	537.07–1652.83
RWavEnLL_s-6	0.2454	0.1586	1408.96	786.48–2145.92	1142.90	742.80–1637.12
RMean	0.2444	0.1067	31.58	25.23–47.44	28.54	19.63–35.84
RCZ4D6SumAverg	0.2439	0.0973	16.50	13.01–24.54	14.80	10.57–18.86
**Ki-67 Negative vs. Ki-67 Positive**						
**Texture Parameters**	**F**	***p*-Value**	**Ki-67 Negative**		**Ki-67 Positive**	
			**Median**	**IQR**	**Median**	**IQR**
RWavEnLH_s-6	0.17	0.1446	42.28	27.07–56.67	32.77	15.98–50.05
RATeta4	0.1309	0.1446	0.16	0.12–0.17	0.14	0.12–0.16
RCN1D6AngScMom	0.129	0.1446	0.01	0.009–0.04	0.02	0.01–0.04
RCV1D6AngScMom	0.129	0.1574	0.01	0.01–0.04	0.02	0.01–0.04
RCZ1D6AngScMom	0.1284	0.1477	0.01	0.009–0.04	0.02	0.01–0.04
RCV2D6AngScMom	0.126	0.1477	0.01	0.006–0.02	0.01	0.008–0.029
RCN2D6AngScMom	0.1246	0.1509	0.01	0.006–0.02	0.01	0.008–0.02
RCH2D6AngScMom	0.1244	0.1856	0.02	0.01–0.04	0.03	0.01–0.04
RCH3D6AngScMom	0.1244	0.1675	0.01	0.008–0.03	0.02	0.01–0.03
RCV3D6AngScMom	0.1242	0.1509	0.01	0.005–0.02	0.01	0.007–0.02
**HER2 Negative vs. HER2 Positive**						
**Texture Parameters**	**F**	***p*-Value**	**HER2 Negative**		**HER2 Positive**	
			**Median**	**IQR**	**Median**	**IQR**
RWavEnLH_s-6	0.7015	0.0037	40.78	26.56–56.72	22.20	8.96–36.05
RWavEnHH_s-4	0.6693	0.0111	9.83	6.51–13.32	6.30	2.98–8.67
RRZD6ShrtREmp	0.5754	0.0457	0.79	0.73–0.82	0.72	0.68–0.80
RRHD6ShrtREmp	0.5677	0.0729	0.67	0.59–0.70	0.61	0.52–0.66
RRND6ShrtREmp	0.5622	0.0524	0.79	0.73–0.82	0.72	0.68–0.80
RRVD6ShrtREmp	0.5377	0.0706	0.77	0.72–0.80	0.70	0.65–0.78
RRHD6Fraction	0.5063	0.0357	0.49	0.41–0.54	0.40	0.34–0.48
RCH1D6InvDfMom	0.4597	0.0426	0.72	0.68–0.77	0.77	0.71–0.81
RCH2D6InvDfMom	0.4571	0.037	0.56	0.51–0.64	0.63	0.57–0.69
RCH3D6InvDfMom	0.4474	0.0332	0.47	0.42–0.55	0.55	0.48–0.61

IQR, interquartile range.

**Table 4 cancers-14-03905-t004:** Multivariate analysis of the parameters independently associated with the HER2+ status.

Independent Variables	Coefficient	Std. Error	*t*	*p*	r_partial_	r_semipartial_	VIF
RCH1D6InvDfMom	12.71	19.76	0.64	0.522	0.08	0.07	813.42
RCH2D6InvDfMom	−43.60	38.37	−1.13	0.260	−0.14	0.12	6252.58
RCH3D6InvDfMom	10.69	20.28	0.52	0.599	0.06	0.05	2153.36
RRHD6Fraction	−19.03	9.60	−1.98	0.052	−0.24	0.22	499
RRZD6ShrtREmp	−1.61	2	−0.8	0.424	−0.1	0.08	10.73
RWavEnHH_s_4	−0.02	0.01	−2.05	**0.044**	−0.24	0.22	2.33
RWavEnLH_s_6	−0.002	0.001	−1.09	0.279	−0.13	0.12	1.46

*t*-value; *p*-value; VIF, variance inflation factor.

**Table 5 cancers-14-03905-t005:** Parameters not independently associated with the HER2 status. Between the brackets are the values corresponding to the 95% confidence interval.

Parameter	*p*-Value	AUC	J	Cut-Off	Se (%)	Sp (%)
RWavEnLH_s_6	0.0002	0.759 (0.644–0.852)	0.3977	≤42.41	92.31 (64.0–99.8)	47.46 (34.3–60.9)
RRZD6ShrtREmp	0.0344	0.678 (0.557–0.783)	0.3690	≤0.72	53.85 (25.1–80.8)	83.05 (71.0–91.6)
RRHD6Fraction	0.0212	0.687 (0.567–0.791)	0.3716	≤0.48	84.62 (54.6–98.1)	52.54 (39.1–65.7)
RCH1D6InvDfMom	0.0249	0.681 (0.560–0.786)	0.3977	>0.706	92.31 (64.0–99.8)	47.46 (34.3–60.9)
RCH2D6InvDfMom	0.0211	0.686 (0.566–0.790)	0.3977	>0.549	92.31 (64.0–99.8)	47.46 (34.3–60.9)
RCH3D6InvDfMom	0.0174	0.690 (0.570–0.794)	0.3716	>0.478	84.62 (54.6–98.1)	52.54 (39.1–65.7)

AUC—area under the curve; Se—sensitivity; Sp—specificity

**Table 6 cancers-14-03905-t006:** Multivariate analysis of the parameters independently associated with the PR+ status.

Independent Variables	Coefficient	Std. Error	*t*	*p*	r_partial_	r_semipartial_	VIF
RPerc01	−0.01	0.02	−0.62	0.5329	−0.07	0.07	5.8
RPerc10	−0.001	0.01	−0.08	0.9296	−0.01	0.01	6.33
RWavEnLH_s_4	−0.004	0.003	−1.17	0.2445	−0.14	0.13	1.44

Std.—standard; *t*-value; *p*-value; VIF, variance inflation factor.

## Data Availability

The data are available only by request.

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
