# Peer review of "Radiomic Signatures Derived from Hybrid Contrast-Enhanced Ultrasound Images (CEUS) for the Assessment of Histological Characteristics of Breast Cancer: A Pilot Study"

_cancers, 2022, doi:10.3390/cancers14163905_

Round 1
Reviewer 1 Report
Authors cleared that hybrid CEUS radiomic features allow the discrimination between breast cancers of different receptor and HER2 status with high specificity. This method can expect; 1) the potential to provide a noninvasive, 2) easily accessible and 3) contrast-agent-safe method to assess tumor biology before and during treatment.
Authors should indicated that the hybrid CEUS radiomic features ensure that the results are as described in the article, regardless of which radiologist performs the diagnosis.
Authors should indicate that the advantages of performing hybrid CEUS radiomic features compared to existing methods in terms of time and cost of measurement.
Author Response
Dear Reviewer,
Thank you for your time and effort spend reviewing our manuscript. In the following, we will try to give an eloquent answer to your observations.
Reviewer 1:
Authors cleared that hybrid CEUS radiomic features allow the discrimination between breast cancers of different receptor and HER2 status with high specificity. This method can expect; 1) the potential to provide a noninvasive, 2) easily accessible and 3) contrast-agent-safe method to assess tumor biology before and during treatment.
Point 1: Authors should indicated that the hybrid CEUS radiomic features ensure that the results are as described in the article, regardless of which radiologist performs the diagnosis.
Answer 1: Thank you for the very good observation. We added in the manuscript the following statement (Lines 143-148)
“To ensure the repeatability and the reproducibility of our results, in the first step of our texture analysis process, the images were normalized. Therefore, we excluded the grey-level variations that could affect the true textures of the images. In the second step, the segmentation process was performed semi-automatically, based on gradient and geometry coordinates, therefore lowering the variability produced by the manual segmentation.“
Point 2: Authors should indicate that the advantages of performing hybrid CEUS radiomic features compared to existing methods in terms of time and cost of measurement.
Thank you for your remark. We added in the manuscript the following statement (Lines 370-375)
“CEUS is an imaging technique comparable in terms of cost and performance time to CESM, but it involves less time and lower costs compared to DCE-MRI. This new approach, which involves radiomic analysis of hybrid CEUS images has additional costs to CEUS the costs related to the MaZda program. Furthermore, it would be interesting to conduct a study in order to explore the learning curve of this method, although we predict that it is similar to the ones involving other imaging techniques.”
____________________________________________________________________________________
We wish to warmly thank you for your expert, thoughtful, and very pertinent observations, which made us realize the omissions we made and, we estimate, greatly helped us improve the paper.
With gratitude,
The Authors

Reviewer 2 Report
This article introduces the potential use of Contrast-enhanced ultrasound (CEUS). The article tested CEUS in 72 patients with known breast cancer, hypothesising that CEUS can give an indication of the biological subtype of breast cancer without an invasive biopsy. This is an innovative non-invasive approach that can be valuable in breast cancer diagnosis. It can be used in larger tumours and those in which significant heterogeneity is thought to exist. I might add that it might also be useful in micro-targeting some tumour areas for biopsy where there is discordance between variable investigative tests' results.
Considering the small sample size and the retrospective nature of the diagnosis, in reality, the study ranks as a CEUS feasibility study. While the authors successfully used CEUS in these patients essentially proving its feasibility. The relatively low sensitivity compared to conventional existing tests is a major limitation of this technique. Whilst the author suggests it's non-invasive, using contrast is not to be taken lightly, especially when these patients usually require multiple diagnostic injections like radionuclear material, patent blue, and various contrast agents if CT or MRI are required.
Innovation and research are key in cancer care development. However, it's absolutely necessary to discuss learning curves and cost-effectiveness when introducing such a new technology. The authors mention that the CEUS was done by a single radiologist which eliminates inter-operator variability, which is an inherent factor in any US study. Do the authors' have any insight on how this might impact the application of CEUS in breast cancer patients?
I disagree with the authors in their claim that CEUS might be an alternative to MRI. The modalities are different and so are their uses and indications. Furthermore, they suggested that CEUS can help differentiate between benign versus malignant solid lesions, which in my opinion is not accurate given that this is outside the scope of this study and was never actually tested.
Given the experimental nature of this study, more details about ethical approval should be reported.
Author Response
Dear Reviewer,
Thank you for your time and effort spend reviewing our manuscript. In the following, we will try to give an eloquent answer to your observations.
Reviewer 2:
This article introduces the potential use of Contrast-enhanced ultrasound (CEUS). The article tested CEUS in 72 patients with known breast cancer, hypothesising that CEUS can give an indication of the biological subtype of breast cancer without an invasive biopsy. This is an innovative non-invasive approach that can be valuable in breast cancer diagnosis. It can be used in larger tumours and those in which significant heterogeneity is thought to exist. I might add that it might also be useful in micro-targeting some tumour areas for biopsy where there is discordance between variable investigative tests' results.
Point 1: Considering the small sample size and the retrospective nature of the diagnosis, in reality, the study ranks as a CEUS feasibility study. While the authors successfully used CEUS in these patients essentially proving its feasibility. The relatively low sensitivity compared to conventional existing tests is a major limitation of this technique. Whilst the author suggests it's non-invasive, using contrast is not to be taken lightly, especially when these patients usually require multiple diagnostic injections like radionuclear material, patent blue, and various contrast agents if CT or MRI are required.
Answer 1: Thank you for the very good observation. Indeed, the relatively low sensitivity of our method needs to be acknowledged as a limitation. We mentioned this section in the “Limitations” section at the end of the manuscript. (Lines 381-385)
“The relatively low sensitivity compared to conventional existing tests is a major limitation of our method. However, this method could still be used as an additional criterion to increase the diagnostic confidence, especially since the technique required for the extraction of the TA parameters is relatively straightforward.”
Point 2: Innovation and research are key in cancer care development. However, it's absolutely necessary to discuss learning curves and cost-effectiveness when introducing such a new technology.
Answer 2: Thank you for the pertinent question. We added in the manuscript the following statements: (Lines 370-375)
“CEUS is an imaging technique comparable in terms of cost and performance time to CESM, but it involves less time and lower costs compared to DCE-MRI. This new approach which involves radiomic analysis of hybrid CEUS images has additional costs to CEUS the costs related to the MaZda program. Furthermore, it would be interesting to conduct a study in order to explore the learning curve of this method, although we predict that it is similar to the ones involving other imaging techniques.”
Point 3: The authors mention that the CEUS was done by a single radiologist which eliminates inter-operator variability, which is an inherent factor in any US study. Do the authors' have any insight on how this might impact the application of CEUS in breast cancer patients?
Answer 3: Thank you for the pertinent question. Indeed, CEUS was done by a single radiologist which eliminates inter-operator variability and it needs to be acknowledged as a limitation. We mentioned this section in the “Limitations” section at the end of the manuscript. (Lines 385-388)
“All CEUS examinations were performed by a single radiologist, which eliminated inter-operator variability, but we used the same parameters, and thus by following the protocol the method could be reproducible.”
Point 3. I disagree with the authors in their claim that CEUS might be an alternative to MRI. The modalities are different and so are their uses and indications. Furthermore, they suggested that CEUS can help differentiate between benign versus malignant solid lesions, which in my opinion is not accurate given that this is outside the scope of this study and was never actually tested.
Answer 3: Thank you for the very good observation. We performed the necessary adjustments by deleting the statement considering that it was not the purpose of our study.
Point 4: Given the experimental nature of this study, more details about ethical approval should be reported.
Answer 4: Thank you for your remark. We added in the manuscript the following statement: (Lines 84-87)
”The informed consent included information related to the purpose and duration of the study, risks, benefits, responsibility of the participant, the rights of the participant to withdraw from the study, confidentiality and the procedure, including the way of administration of the contrast agent and contraindications.”
____________________________________________________________________________________
We wish to warmly thank you for your expert, thoughtful, and very pertinent observations, which made us realize the omissions we made and, we estimate, greatly helped us improve the paper.
With gratitude,
The Authors

Reviewer 3 Report
This manuscript reports the investigators' preliminary methods and results for using radiomics as a noninvasive method to characterize breast cancer histopathological findings from CEUS images. The results are interesting, however, the selection of subjects excluded benign biopsy-proven lesions and malignant lesions under 6mm diameter. While WavEnHH_s_4 parameter was found to have relatively high sensitivity and specificity predicting HER2+ status, I think it is also important to verify that these predictions will not produce false positive results in benign tumors.
As pointed out by the authors, the study numbers are relatively small from a single institution. Would these radiomic results also apply to other institutions using slightly different CEUS imaging protocols? Address how this might influence the results of the study.
Author Response
Dear Reviewer,
Thank you for your time and effort spend reviewing our manuscript.In the following, we will try to give an eloquent answer to your observations.
Reviewer 3:
This manuscript reports the investigators' preliminary methods and results for using radiomics as a noninvasive method to characterize breast cancer histopathological findings from CEUS images.
Point 1:The results are interesting, however, the selection of subjects excluded benign biopsy-proven lesions and malignant lesions under 6mm diameter. While WavEnHH_s_4 parameter was found to have relatively high sensitivity and specificity predicting HER2+ status, I think it is also important to verify that these predictions will not produce false positive results in benign tumors.
Answer 1: Thank you for your remark. We added in the manuscript the following statement: (Lines 326-328)
“It would be interesting to conduct a study in order to investigate if this parameter could predict HER2+ status in benign lesions, thus producing false positive results.”
Point 2:As pointed out by the authors, the study numbers are relatively small from a single institution. Would these radiomic results also apply to other institutions using slightly different CEUS imaging protocols? Address how this might influence the results of the study.
Answer 2: Thank you for your remark. We added in the manuscript the following statement: (Lines 378-380)
“but it would be interesting to evaluate a larger database, with examinations performed in several institutions, even if slightly different CEUS imaging protocols were used, to compare the results and evaluate the reproducibility of the method”
____________________________________________________________________________________
We wish to warmly thank you for your expert, thoughtful, and very pertinent observations, which made us realize the omissions we made and, we estimate, greatly helped us improve the paper.
With gratitude,
The Authors

Reviewer 4 Report
Based on the hypothesis of spatial heterogeneity of breast cancer, the authors have investigated the feasibility of differentiating immunohistochemical characteristics by texture features extracted from CEUS images. Feature selection, significant test, and multivariate analysis were used for feature number reduction. Overall, the study method and results are clearly presented.
Major points:
1. What are the advantages or benefits of using radiomic features over other quantitative features? Though many advantages of radiomics have been discussed (in lines 259-261, 323-324), the study motivation can be more justified if the authors can provide some comparisons between radiomic features and other quantitative features such as contrast enhancements characteristics used in another study (https://www.ncbi.nlm.nih.gov/pmc/articles/PMC5453558/).
2. It is nice to see some comparisons with related studies in the discussion section. However, some of them are more suitable to be considered as backgrounds and it would be more appropriate to include them in the introduction section. For instance, contents in lines 259-275 can be used for justifying the motivation of this study and they might be better to be placed before the materials and methods section.
3. Names of the mentioned texture parameters are not defined. What's the reason for adding the letter "R" ahead in all feature names? Why not directly name the wavelet parameter as "WavEnLH" rather than "RWavEnLH"? For example, in "RCV4D6DifVarnc", "DifVarnc" is defined in Table 1 but the meaning of "RCV4D6" is not defined in the manuscript.
4. What does the second limitation in line 346 mean? It looks molecular types are fine but only several pathological types have very few samples in Table 2.
5. What are the 5 scales and 4 frequency bands used for the wavelet transform features extraction?
Minor points:
6. Line 62: Correct "phenotipical" to "phenotypical".
7. Table 2: The PR- cases should be 43 (assuming the percentage of 59.73% is correct) instead of 38.
8. Use full names, such as "Interquartile range" for IQR at its first appearance.
9. Use "indicates" in line 352.
10. Will the word "present" be better than "represent" in line 21?
11. Double-check the language style. For example, "utilised" used in line 349 is British English and "color" in line 103 is American English.
12. There is an empty row in Table 1.
Author Response
Response to Reviewer’s Comments
Dear Reviewer,
Thank you for your time and effort spend reviewing our manuscript.In the following, we will try to give an eloquent answer to your observations.
Reviewer 4:
Based on the hypothesis of spatial heterogeneity of breast cancer, the authors have investigated the feasibility of differentiating immunohistochemical characteristics by texture features extracted from CEUS images. Feature selection, significant test, and multivariate analysis were used for feature number reduction. Overall, the study method and results are clearly presented.
Major points:
Point 1.What are the advantages or benefits of using radiomic features over other quantitative features? Though many advantages of radiomics have been discussed (in lines 259-261, 323-324), the study motivation can be more justified if the authors can provide some comparisons between radiomic features and other quantitative features such as contrast enhancements characteristics used in another study (https://www.ncbi.nlm.nih.gov/pmc/articles/PMC5453558/).
Answear 3: Thank you for your suggestion. We added in the manuscript the following statement (Lines 86-89)
“Furthermore, Wang et al. (17) found a correlation between CEUS and acoustic radiation force impulse (ARFI), which also provides quantitative features, regarding the assessment of HER-2 expression levels in breast cancer.”
Point 2.It is nice to see some comparisons with related studies in the discussion section. However, some of them are more suitable to be considered as backgrounds and it would be more appropriate to include them in the introduction section. For instance, contents in lines 259-275 can be used for justifying the motivation of this study and they might be better to be placed before the materials and methods section.
Answear 2: Thank you for your suggestion. We reorganized the manuscript by moving the indicated paragraph in the introduction section (Lines 72-86)
Point 3.Names of the mentioned texture parameters are not defined. What's the reason for adding the letter "R" ahead in all feature names? Why not directly name the wavelet parameter as "WavEnLH" rather than "RWavEnLH"? For example, in "RCV4D6DifVarnc", "DifVarnc" is defined in Table 1 but the meaning of "RCV4D6" is not defined in the manuscript.
Answear 3:Thank you for the very good observation. We added in the manuscript the following statement: (Lines 214-218)
“Abbreviations of feature characteristics produced by the extraction algorithm appear in the feature name generated by MaZda software. The first letter indicates the color channel through which the image was computed (R” identities the red color channel). This color channel was automatically chosen by the software, the investigators had no intervention on this procedure.”
Point 4.What does the second limitation in line 346 mean? It looks molecular types are fine but only several pathological types have very few samples in Table 2.
Answear 4:Thank you for the question. We wanted to point out that triple-negative and HER2+ molecular subtypes were poorly represented (7, respectively 6 cases).
We modified the statement “Second, some molecular subtypes (HER2+ and triple-negative) are too poorly represented” (Line 405)
Point 5.What are the 5 scales and 4 frequency bands used for the wavelet transform features extraction?
Answear 5:Thank you for the question.
The Wavelet transformation process consists of three steps. In the first step, images are scaled up five times both in vertical and horizontal directions. Furthermore, two types of filters (high and low pass) are applied to separate the image data. Finally, different subbands are extracted from an image that becomes subdivided into four parts, corresponding to different frequency components. The result is a five-scaled image with four frequency bands on every scale, each labeled as LL (low-low), HL (high-low), LH (low-high), and HH (high-high). The LL band contains low-frequency signal contents, whereas the HH band contains high-frequency signal contents.
The explanation can be found in the manuscript (Lines 313-321). If you consider it appropriate, we can rephrase it.
Minor points:
Point 6. Line 62: Correct "phenotipical" to "phenotypical".
Answear 6:Thank you for the very good observation. We corrected the word in the manuscript. (Line 66)
Point 7.Table 2: The PR- cases should be 43 (assuming the percentage of 59.73% is correct) instead of 38.
Answear 7:Thank you for the very good observation. We corrected the number of cases in Table 2, we had 34 PR- cases and 38 PR+ cases.
Point 8. Use full names, such as "Interquartile range" for IQR at its first appearance.
Answear 8:Thank you for the very good observation. We added the explanation of IQR under Table 3. (Line 246)
Point 9.Use "indicates" in line 352.
Answear 9:Thank you for the suggestion. We corrected the word in the manuscript. (Line 379)
Point 10.Will the word "present" be better than "represent" in line 21?
Answear 10:Thank you for the suggestion. We corrected the word in the manuscript.(Line 21)
Point 11.Double-check the language style. For example, "utilised" used in line 349 is British English and "color" in line 103 is American English.
Answear 11:Thank you for the very good observation. We corrected the word "utilised" in the manuscript.(Line 414). We also reviewed the manuscript for other similar mistakes.
Point 12.There is an empty row in Table 1.
Answear 12:Thank you for the very good observation. We deleted the empty row.
____________________________________________________________________________________
We wish to warmly thank you for your expert, thoughtful, and very pertinent observations, which made us realize the omissions we made and, we estimate, greatly helped us improve the paper.
With gratitude,
The Authors

Round 2
Reviewer 2 Report
Many thanks to the authors for your review. The study introduces an innovative technique which promises great potential.
Can the authors elaborate more on the contrast used, please? More details on the exact drug, dosing, and timing of the contrast in relation to performing the US, possible side effects, and whether any patients suffered any morbidity from it during the study.
Author Response
Response to Reviewer’s Comments
Dear Reviewer,
Thank you for your time and effort spend reviewing our manuscript.In the following, we will try to give an eloquent answer to your observations.
Many thanks to the authors for your review. The study introduces an innovative technique which promises great potential.
Point 1:
Can the authors elaborate more on the contrast used, please? More details on the exact drug, dosing, and timing of the contrast in relation to performing the US, possible side effects, and whether any patients suffered any morbidity from it during the study.
Answer 1:Thank you for your suggestion. We added in the body text the suggested information: (Lines 132-133, 140-143)
For all the examinations we set the mechanical index (MI) at 0.06, the gain of 100–120 dB, single focus, image depth of approximately 3–4 cm, and SonoVue®(sodium hexachloride microbubbles; Bracco, Amsterdam, Netherlands) was administered intravenously as a bolus of 2.4ml followed by a flush of 5mL sodium chloride 0.9% using a 20 gauge cannula.
We started recording the examination synchronously with the injection of the contrast agent. No patient presented side effects to SonoVue® (sensitivity at the injection site, hypersensitivity reactions such as facial flushing, nausea, bradycardia, hypotension or anaphylactic shock).
____________________________________________________________________________________
We wish to warmly thank you for your expert, thoughtful, and very pertinent observations, which made us realize the omissions we made and, we estimate, greatly helped us improve the paper.
With gratitude,
The Authors